# Spatial constraints govern competition of mutant clones in human epidermis

M.D. Lynch[1,2], C.N.S. Lynch[1], E. Craythorne[2], K. Liakath-Ali[1], R. Mallipeddi[2], J.N. Barker[2] & F.M. Watt[1]

Deep sequencing can detect somatic DNA mutations in tissues permitting inference of clonal relationships. This has been applied to human epidermis, where sun exposure leads to the accumulation of mutations and an increased risk of skin cancer. However, previous studies have yielded conflicting conclusions about the relative importance of positive selection and neutral drift in clonal evolution. Here, we sequenced larger areas of skin than previously, focusing on cancer-prone skin spanning five decades of life. The mutant clones identified were too large to be accounted for solely by neutral drift. Rather, using mathematical modelling and computational lattice-based simulations, we show that observed clone size distributions can be explained by a combination of neutral drift and stochastic nucleation of mutations at the boundary of expanding mutant clones that have a competitive advantage. These findings demonstrate that spatial context and cell competition cooperate to determine the fate of a mutant stem cell.

[1] Centre for Stem Cells and Regenerative Medicine, King's College London, London SE1 9RT, UK. [2] St John's Institute of Dermatology, King's College London, London SE1 9RT, UK. Correspondence and requests for materials should be addressed to F.M.W. (email: fiona.watt@kcl.ac.uk)

In mice, the use of genetic lineage tracing is a well-established technique for identifying subpopulations of cells that contribute to tissue homeostasis and disease[1]. Typically, a specific or ubiquitous gene promoter is used to express Cre recombinase in the cells of interest and their progeny are fluorescently labelled for analysis. In human tissues, however, cell relationships must be inferred by other approaches. Historically, these have included the use of spontaneous mutations in mitochondrial and genomic DNA as clonal markers, in combination with analysis of methylation patterns in non-expressed genes[2, 3]. More recently, deep sequencing has allowed the detection of hundreds of mutated genes and is being widely used to infer clonal relationships in a variety of tumour types[4, 5]. One human tissue that lends itself to clonal analysis is the outer covering of the skin, the epidermis. The epidermis is maintained by cells that self-renew in the basal layer and differentiate in the suprabasal layers, forming a stratified squamous epithelium[6]. Skin is readily accessible in the form of surgical waste, and the techniques for whole-mount epidermal immunolabelling are well established[7]. Furthermore, the risk of skin cancer increases exponentially with age and is associated with accumulation of somatic mutations[8]. Genes that are frequently mutated in cutaneous squamous cell[9] and basal cell[10] carcinoma have been identified and can be used to infer clonal relationships. However, previous studies reveal a paradox, whereby there is evidence of positive selection of mutant epidermal clones[11], yet clone size distributions are consistent with neutral drift[12–14], a process by which the emergence of mutant clones is through genetic drift of mutant alleles that have neither a positive nor a negative effect on clone size.

One potential solution to this paradox is that there is competition between mutant cells. Cell competition is an evolutionarily conserved mechanism that leads to the outgrowth or elimination of relatively less fit cells from a tissue by competition with fitter cells. It was initially described in the developing Drosophila epithelium, where mutant cells are at a competitive disadvantage[15]. Subsequently it was demonstrated that mutant cells can have a competitive advantage over neighbouring cells[16] and that cell competition can play a physiological role in the regulation of cell populations[17–19]. We hypothesised that a similar mechanism may contribute to the differential survival and proliferation of mutant clones in the epidermis.

Here we reasoned that our understanding of clonal relationships and the potential role of cell competition in sun-exposed human skin could be improved by analysing more and larger samples than previously, by extending the analysis to skin from older individuals, and by sampling skin from donors who were at elevated risk of developing skin cancer. These approaches have led us to discover that clone size cannot be explained solely on the basis of neutral drift, but is also influenced by the spatial location of cells that acquire secondary mutations.

## Results

### Identification of mutations in cancer-prone skin

We obtained epidermis and matched genomic (salivary) samples from 10 patients aged 33–87 undergoing Mohs micrographic surgery for non-melanoma skin cancer[20] (Supplementary Fig. 1a–c). During this procedure, thin layers of cancer-containing skin are progressively removed from the margin of the tumour and until only cancer-free tissue remains. The risk of subsequent skin cancer is substantially increased in individuals who have already had a tumour excised[21]. Samples for sequencing were obtained from excess skin removed from the clear margin adjacent to the tumour at the time of reconstruction and were trimmed to give a total skin surface area of 16 mm$^2$ per patient for DNA extraction. This is a 16-fold greater area than sequenced in earlier studies.

A capture oligonucleotide strategy was designed to target 121 genes frequently mutated in cutaneous squamous cell[9] and basal cell[10] carcinoma (Fig. 1a). We identified a total of 887 somatic mutations across the 10 epidermal samples analysed (Fig. 1b; Supplementary Data 1). The mutational spectrum was dominated by C-to-T transitions, a characteristic of ultraviolet light-induced mutagenesis[22] (Fig. 1c, d). All patients had individual clones present with a variant allele fraction (VAF) in excess of 0.05 (Fig. 1e, f), equating to 10% of the surface area of each biopsy. There was a non-linear relationship between the number of clones per patient and age (Fig. 1g), with the oldest patient having a cumulative VAF of >5 (Fig. 1f). The largest mutant alleles and the most frequently mutated gene was NOTCH1 (Fig. 2a, b) with mutant NOTCH1 and NOTCH3 clones detected in all 10 patients (Supplementary Fig. 2c). Other genes mutated in large clones include known drivers of epithelial malignancy such as NOTCH2, TP53, KMT2B, KMT2C, and potentially novel regulators of epidermal stem cell dynamics such as PLXNB3 and RAI1 (Supplementary Fig. 2d–j). Genes that are recurrently mutated in cutaneous squamous cell carcinoma are also frequently mutated in sun-exposed epidermis (Fig. 2c) and the location of mutations within the NOTCH1 gene in our samples closely parallels that seen in cutaneous squamous cell carcinoma (Fig. 2d, e; Supplementary Fig. 3).

### Analysis of clone size distributions

The stochastic division and differentiation of stem cells within the basal layer of the epidermis gives rise to a process of neutral drift, whereby a minority of clones expand and others are extinguished[23, 24]. This leads to a characteristic distribution of clone sizes[12]. In our samples, the shape of the clone size distribution appeared to show a good fit to the neutral drift model both when clone sizes are aggregated across all patients (Fig. 3a, b) and when analysed on a per-patient basis (Supplementary Fig. 4), although there was a small excess of larger clones (VAF > 0.14) with mutations predicted to change protein function but not with neutral SNVs (Fig. 3c–f). However, neutral drift cannot account for the large size of the distribution. Clones that exhibited a VAF of >0.20 corresponded to an area of >6.4 mm$^2$ ~64,000 basal epidermal cells. Clones approaching this size have also been observed by immunohistochemical staining of mutant TP53[7, 25] in human skin. Several independent mathematical analyses indicate that the time required for these larger clones to arise by neutral drift would exceed the maximal human lifespan by an order of magnitude (>1000 years; see 'Methods' section). This implies that certain clones must exhibit a selective advantage and is consistent with mouse models showing that experimental inactivation of the Notch pathway in the basal layer of the mouse oesophagus[26] or hedgehog signalling in mouse epidermal stem cells[27] leads to the expansion of contiguous clones that have a competitive advantage over neighbouring stem cells.

### Simulation of clonal evolution in a homogeneous basal epidermal layer

To understand the large size of the clone size distribution, we developed a lattice model of stem cell behaviour in a homogenous two-dimensional stem cell compartment (Fig. 3g–i; Supplementary Methods). As anticipated, when all stem cells are equivalent, the model reproduces the clone size distribution predicted by neutral drift (Fig. 3j, k; Supplementary Fig. 5). We next modelled the consequence of non-neutral mutations that arise stochastically and reduce the probability of a cell exiting the stem cell compartment (Supplementary Movie 1; Fig. 3l–q; Supplementary Fig. 9). Unexpectedly, for clones detectable by sequencing (VAF > 0.007), the distribution replicates that of neutral drift, but on a larger scale. The shape of this distribution is not dependent upon the frequency of non-neutral

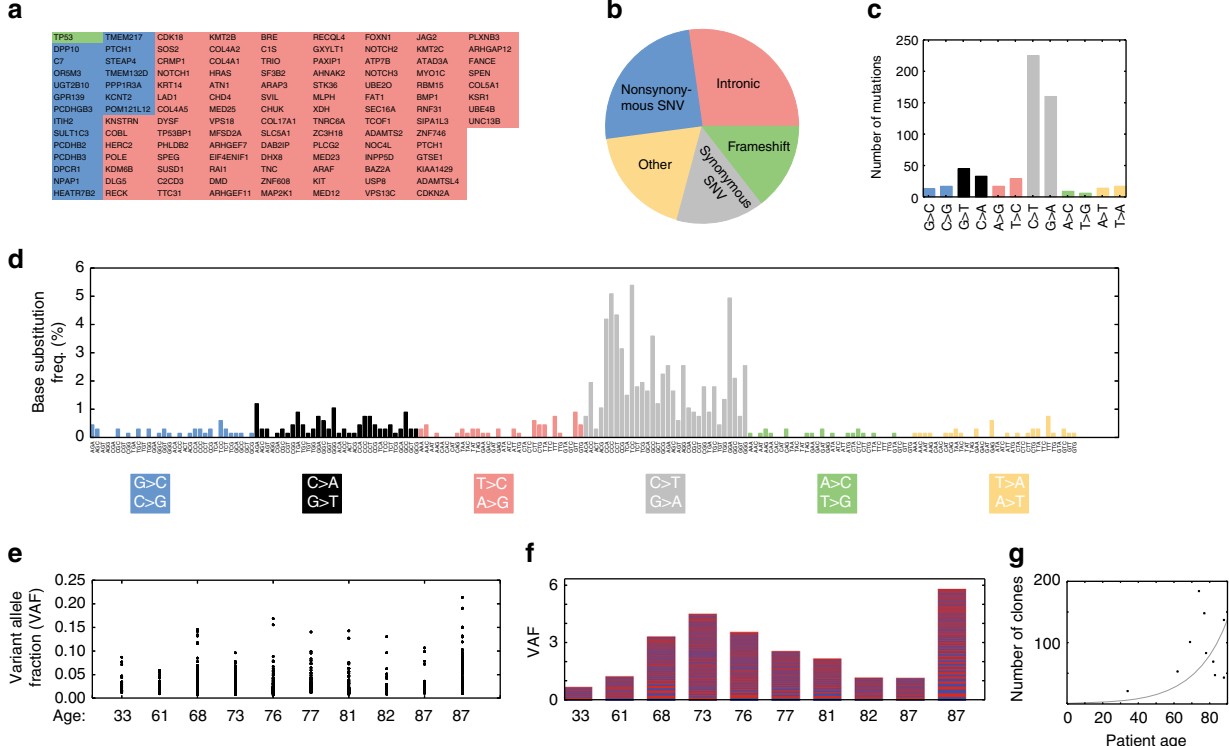

**Fig. 1** Identification of clonally expanded mutations in human epidermis. **a** Identity of genes included in capture oligonucleotide set. Genes frequently mutated in SCC (red) BCC (blue) or both (green) were included. **b** Genomic context of mutations identified by capture sequencing. **c** Number of base substitutions on the coding strand. The dominance of C > T transitions is characteristic of UV light-induced mutagenesis. The excess of C > T over G > A is a characteristic of transcription-coupled repair. **d** Base substitutions plotted according to trinucleotide context on the coding strand. **e** Mutant clone size distribution for each patient. The size of each mutant clone identified (variant allele fraction, VAF) is indicated. Each filled circle indicates a single mutant clone. **f** Cumulative clone size distributions for each patient. Clones are stacked in order of size with largest at the base of the column. Even clones are blue and odd red. **g** Relationship between mutant clone distribution and patient age. Each point represents the number of clones identified for a single patient. An exponential line of best fit is plotted

mutation (Supplementary Figs. 7, 8) or the magnitude of the competitive advantage, but a greater competitive advantage leads to more rapid expansion (Supplementary Fig. 10a–f).

Previous experimental studies[7, 28–30] indicate that the rate of stem cell migration is negligible in the basal layer of the epidermis in the steady state. This is supported by lattice-based modelling, which demonstrates that a non-negligible rate of stem cell migration leads to dispersed, scattered clones (Supplementary Fig. 6) rather than the spatially contiguous clones that are observed experimentally[7, 25–27]. However, even if we assume a high rate of migration (10 μm/day), this does not alter the clone size distribution (Supplementary Fig. 10g–k). Furthermore, relaxation of the spatial constraints to permit more than one cell per lattice point does not significantly alter the distribution (Supplementary Fig. 13a–f) or the shape of mutant clones (Supplementary Fig. 13i–l).

Our observation is simple to understand conceptually (Fig. 3r–t). If a neutral mutation arises in a wild-type stem cell (blue, cyan), it will be subjected to neutral drift and on an average will neither expand nor contract. Similarly, if a neutral mutation arises within the centre of an expanding primary clone (pink, green), it will be equivalent to all surrounding mutant stem cells and will again be subjected to neutral drift with respect to its neighbours. If, however, a neutral mutation arises in a cell at the boundary of the expanding primary clone (yellow, purple), the progeny of that cell can expand, replacing surrounding wild-type stem cells.

A geometric formulation of this model (see 'Methods' section) gives an inverse cubic power law dependency for the distribution

at larger clone sizes. In some circumstances, small clones appear to undergo a transient period of exponential growth[26] before reverting to a slower growth rate. Mathematical analysis (see 'Methods' section) reveals that a model analogous to that described above will also arise for clones growing exponentially, in this case, giving rise to an inverse power law distribution with squared rather than cubic dependency. A comparison of the fit of our data to inverse squared and cubic power laws is shown (Fig. 3u).

**Modelling clonal evolution in a heterogeneous basal cell layer.** Having established that the expansion of mutant stem cell clones in a homogenous cellular compartment comprising only stem cells can reproduce the clone size distribution observed in our patients, we extended the model to a heterogeneous stem cell compartment. This comprises clusters of stem cells surrounded by transit amplifying (TA) cells with limited proliferative potential (Supplementary Fig. 11; see 'Methods' section) and more accurately describes the situation in the basal layer of human epidermis[31]. In human epidermis stem cell clusters expressing high levels of β1-integrins are surrounded by cells with more limited proliferative potential, which have a high likelihood of initiating terminal differentiation and expressing keratin 10 (Fig. 4a–c).

We stochastically introduced mutations that prevent stem cells from differentiating to TA cells and reduce the probability of exiting the stem cell compartment (see 'Methods' section). This leads to the expansion of mutant stem cell clones into areas

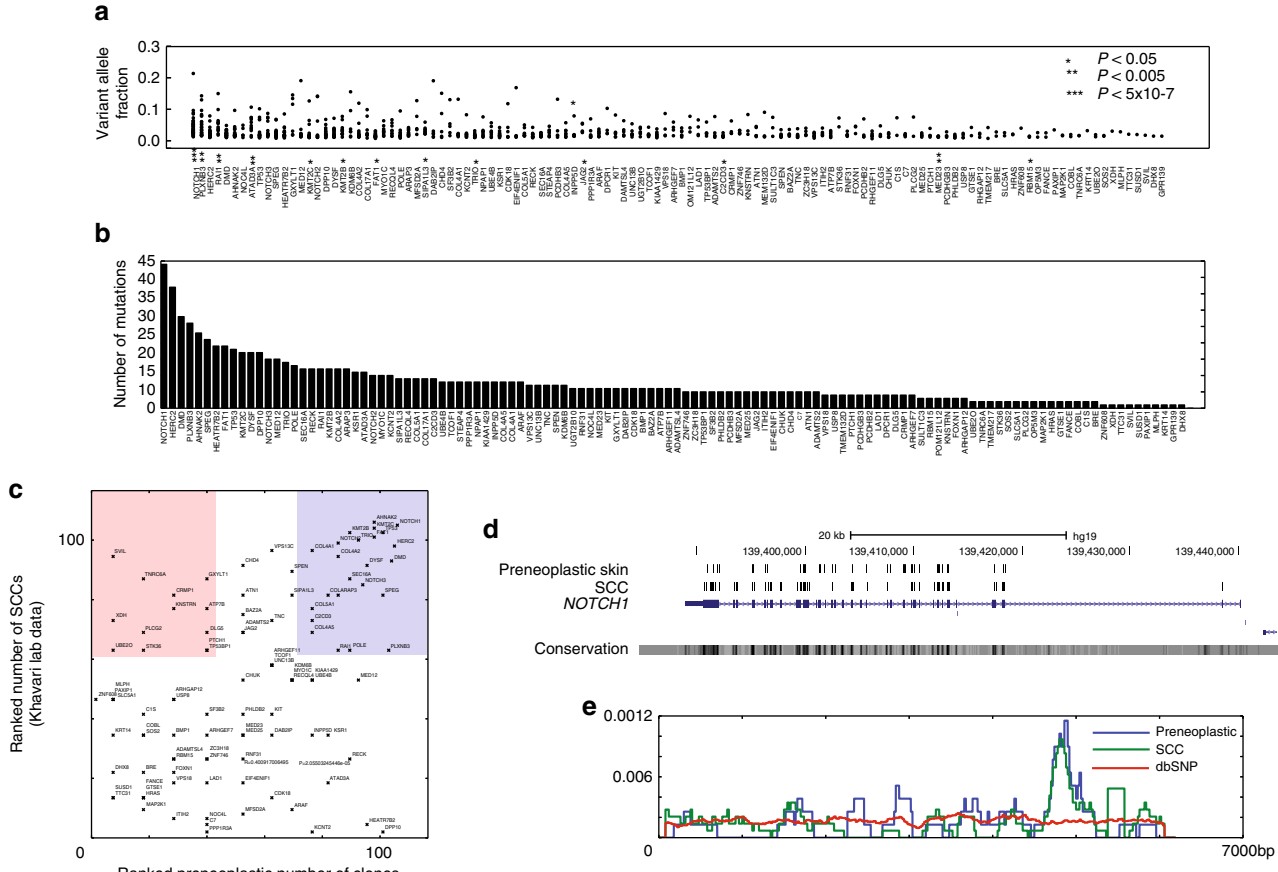

**Fig. 2** Mutational landscape of human epidermis. **a** Clone size distribution for each gene in the capture set plotted according to variant allele fraction (VAF). Each filled circle represents a single identified clone. Significant differences in the shape of the distribution in comparison to the ensemble distribution by the Kolmogorov–Smirnov test are indicated (*). **b** Total number of mutations identified for each gene in the capture set. **c** Comparison of mutation frequency in cutaneous SCC (number of tumours) and preneoplastic epidermis (total number of mutant clones detected across all patients). Each gene is plotted in rank order on both axes. Blue shaded area: mutations prevalent in both cutaneous SCC and preneoplastic epidermis. Red shaded area: mutations prevalent in cutaneous SCC but not in preneoplastic epidermis. **d** Comparison of location of identified mutations in NOTCH1 in preneoplastic epithelium and cutaneous SCCs (Khavari lab data). **e** Density plot illustrating location of mutations within the coding (exonic) sequence of NOTCH1

previously occupied by TA and wild-type stem cells (4d-g; Supplementary Movie 2), with the distribution of mutant clone sizes (Fig. 4i) matching the experimentally observed clone size distribution for both large (Fig. 4j) and small stem cell cluster sizes (Supplementary Fig. 12j). Interestingly, the shape of this distribution is dependent upon the initial spatial arrangement of stem cells, where evolution is determined by neutral drift but not for selective advantage (Supplementary Fig. 12i, j).

This model of mutant clonal expansion in a heterogeneous stem cell compartment is analogous to that observed for disruption of Notch signalling[26]. Mutations affecting the Notch pathway are the most frequent mutations in both our data and previously published data[11]. Activated Notch1 is present in both basal and suprabasal cells of human epidermis (Fig. 4k, l). The clone size distribtion of NOTCH1 mutations in our data is shown (Fig. 4j).

## Discussion

By analysing larger areas of human epidermis, obtaining tissue from individuals of a wide age range, and increasing the number of samples analysed, we have ben able to resolve the current controversy surrounding the evolution of mutant clones[11–14]. We have shown that mutant clone size distributions in epidermis derived from patients at high risk of skin cancer occur on too

large a scale to arise by neutral drift within the timescale of a human lifespan. Through mathematical modelling and computational simulation, we have demonstrated that the observed distribution can be explained by the stochastic fixation of secondary mutations arising at the boundary of expanding clones carrying mutations such as NOTCH1.

Notch signalling is an evolutionarily conserved signalling pathway. Notch1-3 are expressed in the epidermis[32–35] and Notch signalling plays a key role in the regulation of epidermal stem cell self-renewal and differentiation[36]. Notch pathway components are frequently mutated in cutaneous squamous cell carcinoma[9] and other epithelial malignancies[37, 38]. Mouse studies have demonstrated that, in keeping with our observations, stochastically-induced Notch-mutant cells have a competitive advantage, expanding to form large contiguous clones in the mouse oesophagus[26].

While the conclusions of our study are of particular relevance to disruption of Notch signalling, they are potentially applicable to any mechanism that leads to a competitive advantage for the mutant clone. This concept of cell competition was initially recognised in the developing Drosophila wing epithelium[15]. We note that in addition to a reduced rate of loss of mutant cells, a competitive advantage can result from the (non-cell-autonomous) promotion of cell death of adjacent wild-type cells[39]. There is evidence for such a mechanism in the context of

disrupted Notch signalling both in vitro[29] and in vivo[27]. Furthermore, expansion of preneoplastic APC mutant clones in a Drosophila model[40] appears to follow this paradigm. Our experimental data are not able to differentiate the effects of cell autonomous and non-cell autonomous mechanisms, since the consequences for our models are the same, namely a local competitive advantage for cells at the boundary of the expanding clone.

Our conclusion, that the fate of a mutant stem cell is influenced by a combination of neutral drift, cell competition and by the spatial context in which it arises, has important implications for the interpretation of sequencing studies of preneoplastic tissues

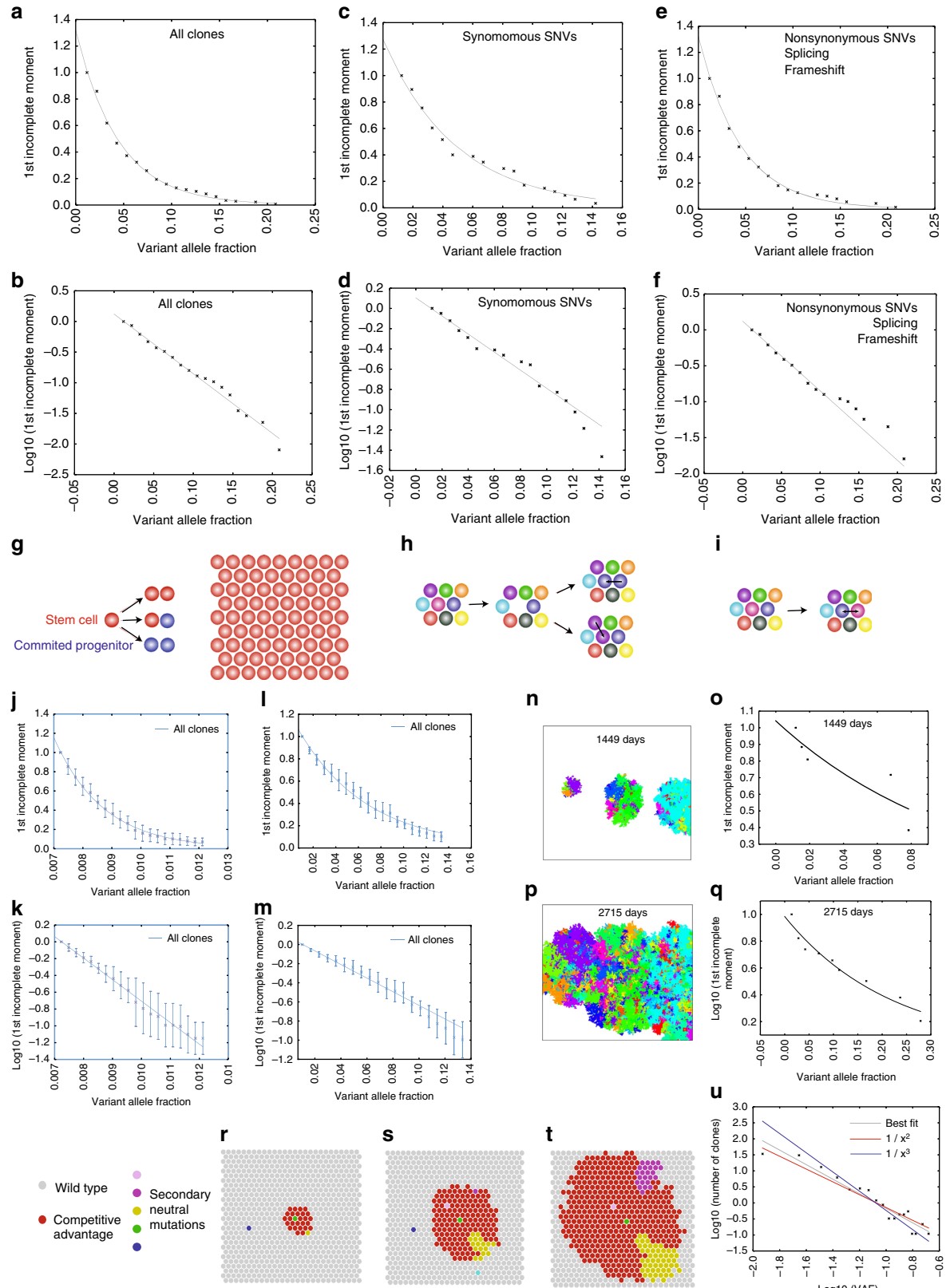

and for understanding the earliest stages in the development of malignancy. It will be interesting to discover whether spatial context is also important in the evolution of mutant clones in other epithelia, such as the intestine, and whether it is a general feature of cancer-associated mutations or is specific to the effects of UV light. We also speculate that, for any given combination of genetic lesions, spatial context will influence the likelihood that they will contribute to tumour progression.

## Methods

**Sample collection and DNA extraction**. Anonymised skin samples samples were obtained from 10 patients of 33–87 years of age with Fitzpatrick skin type I-II undergoing Mohs micrographic surgery for the treatment of non-melanoma skin cancer affecting the head and neck. The project was subject to ethical approval by the research tissue bank at St John's Institute of Dermatology (Guy's and St Thomas' NHS Foundation Trust) and patients gave informed consent for the inclusion of their tissue samples. Samples were collected from excess skin removed at the time of reconstruction of the Mohs defect following histological clearance of the tumour, thus we can be confident that there was no extension of the tumour into the collected skin specimens. Skin samples were flash-frozen and subsequently stored at −80 °C prior to further processing. In addition, salivary samples of 2–3 ml saliva were obtained from these 10 patients for the extraction of genomic control DNA.

Following retrieval from the research tissue bank, fresh frozen skin samples were thawed at room temperature. Samples were trimmed to an area of 4 × 4 mm. It is noted that, irrespective of donor site, skin samples will contract after excision due to innate skin elasticity and there may be differences in the degree of contraction according to donor site. Excess skin was employed for the preparation of paraffin blocks. Haematoxylin and Eosin stains were prepared from all samples. Skin samples from which DNA was to be extracted were incubated at 52 °C for 10 min in phosphate buffered saline (PBS). Following this treatment, the epidermis can readily be separated from the papillary dermis. To ensure that the entire epidermis was collected intact, separation was visualised with a dissecting microscope.

To analyse the large scale clonal architecture of the samples DNA was extracted from the entire 4 × 4 section of epidermis using the Qiagen DNeasy blood and tissue kit according to the manufacturer's instructions. Salivary samples were washed in PBS and subsequently DNA was extracted in an identical manner.

**DNA capture, library preparation and sequencing**. A custom capture strategy (NimbleGen SeqCap EZ) was designed with capture oligos targeting 121 genes frequently mutated in non-melanoma skin cancer (Fig. 1. Although *CDKN2A* was included in the original capture design, capture failed across all samples and therefore it was excluded from all further analysis. Capture was performed according to the SeqCap EZ Library SR User's Guide (Version 4.1; May 2013) and library preparation was performed according to the TruSeq DNA Sample Pre-paration Guide (15005180 Rev. C, June 2011) with no modifications.

Reads were pre-mapped with BWA[41] (parameters: –q10) and subsequently mapped with Stampy[42] run with default parameters then sorted with Picard. Sequencing was performed on the Ilumina Hiseq2000 or Hiseq 25000 with TruSeq SBS v3 chemistry. Average per-sample coverage was 1000×, giving overall rates of coverage across the 10 normal and matched salivary control samples of 20,000×. PCR duplicates and reads with a Phred quality score of <30 were removed from the analysis.

**Identification of significant mutations**. The identification of subclonal somatic mutations in high throughput sequencing data is challenging due to the error rate that reflects a combination of sequencing and PCR artefacts. This error rate varies according to genomic location, and furthermore, whereas a genuine mutation is anticipated to be present equally on both strands, a PCR error will preferentially affect one strand. We employed statistical methodology identical to that reported by Martincorena et al.[11]. Since we did not subdivide the epidermal DNA specimens from each individual, we instead constructed the background error model across all of the control (salivary samples).

For each position in the genome $j$ and each normal salivary DNA samples $i$, $X_{ijk}$, $X'_{ijk}$ denote the count of a specific nucleotide $k \in$ (A, C, T, G, −) with— corresponding to deletion of the base. The background error rate was fit to a beta binomial distribution (Supplementary Fig. 1d):

$$X_{ijk} \sim \mathrm{BetaBin}\left(n_{ij}, v_{ijk}, \rho\right)$$
$$X'_{ijk} \sim \mathrm{BetaBin}\left(n'_{ij}, v'_{ijk}, \rho\right),$$

where $n_{ij}$ and $n'_{ij}$ denote coverage on the forward and reverse strands, respectively, $v_{ijk}$ and $v'_{ijk}$ denote the mean fraction of reads across the normal samples and $\rho$ denotes the overdispersion parameter for the beta binomial fit. As described by Martincorena et al.[11], we estimate a single value for $\rho$ across all the captured regions.

For each genomic position in each epidermal sample, a likelihood ratio test was employed with the null model $H_0$ that observed read counts $x_{jk}$ and $x'_{jk}$ are explained by background beta binomial models $X_{ijk}$ and $X'_{ijk}$, and the alternative model $H_1$ that $x_{jk}$ and $x'jk$ are derived from a beta binomial distribution with mean $\mu_{jk} = \left(x_{jk} + x'_{jk}\right)/\left(n_j + n'_j\right)$. For each base that differed from the reference base, $P$ value was calculated from the likelihood ratio using $\chi^2$ distribution with 2 degrees of freedom. $P$ values were corrected for multiple testing according to the Benjamini–Hochberg method with a $q$ value cutoff of 0.05. In order to eliminate contamination from genomic SNPs, we called subclonal variants separately for salivary and epidermal samples from each patient and variants, which were also present in the matched salivary sample were eliminated from the analysis. It is noted that this analysis will not identify copy number variants.

To further eliminate artifactual clones, we required that genomic locations were covered by a total of >50 reads in both the sample and control samples. We additionally required that mutant clones were supported by a minimum of 5 reads. Finally, we noticed that, where mutations occurred within 5 bases of one another in the same sample, these were typically had similar VAFs and were associated with complex deletion events. These mutations were merged and reported as a single deletion spanning all bases with VAF taken as the average of the individual clones. Statistical methods were validated by a test of sensitivity and false discovery rate on simulated mutations generated via a Monte Carlo simulation (Supplementary Fig. 1f, g) and were implemented in Python. The VGAM R package was employed for fitting of beta binomial distributions.

**Histology and microscopy**. Frozen section staining of human epidermis for activated (cleaved) Notch1 was performed with a commercially available antibody (Abcam, ab8925 1:200). Tissue samples were embedded in optimal cutting temperature compound (Life Technologies) directly or fixed first with 4% paraformaldehyde for 20 min at room temperature, and stored at −80 °C. Ten to sixteen micron sections were cut using a Thermo Cryostat Nx70 (Thermo Fisher Scientific), Sections were transferred to blocking buffer for 1 h at room temperature, labelled with primary antibodies diluted in blocking buffer overnight at 4 °C, washed in PBS and then labelled with secondary antibodies and DAPI for 1 h at room temperature. Fluorescence mounting medium (DAKO) was used for mounting the coverslips (SLS) on the slides.

**Fig. 3** Mutant clone size distributions in human epidermis. **a**, **b** Clone size distribution for all mutations identified in the 10 patients in the study. Unnormalised variant allele fraction (VAF) is plotted against the first incomplete moment on standard (**c**) and log (**d**) scale. **c**, **d** Clone size distribution for synonymous SNVs. Unnormalised variant allele fraction (VAF) is plotted against the first incomplete moment on standard (**e**) and log (**f**) scale. **e**, **f** Clone size distribution for nonsynonymous SNVs, frameshift mutations and splicing mutations. Unnormalised variant allele fraction (VAF) is plotted against the first incomplete moment on standard (**e**) and log (**f**) scale. **g–i** Hexagonal lattice model of stem cell dynamics in the basal layer of the epidermis (**g**). Stem cells are stochastically lost and replaced by division of a neighbouring stem cell (**h**). Cellular migration is implemented by stochastically swapping the position of adjacent cells on the lattice (**i**). **j**, **k** Computational lattice simulation of neutral drift (3 years, 200 × 200 lattice, neutral mutation rate 10−3 cell-1 day-1. Ten replications with mean ± S.D.). First incomplete moment is plotted against VAF on (**j**), standard and (**k**), log scale. **l**, **m** Computational lattice simulation of selective advantage (10 years, 200 × 200 lattice, neutral mutation rate 10−3 cell-1 day-1, non-neutral mutation rate 10−6 cell-1 day-1, 10 replications with clones detectable by sequencing (>0.007 VAF) mean ± S.D.). First incomplete moment is plotted against VAF on (**l**) standard and (**m**) log scale. **n**, **o** Visualisation of a single run of the lattice simulation with non-neutral mutations at (**f**) 1449 and (**h**) 2715 days. Wild-type stem cells are white (not visible). Stem cells carrying a non-neutral mutation are assigned an arbitrary colour with subclones of alternative colour reflecting the acquisition of secondary neutral mutations. **p**, **q** Clone size distributions for a single run of the lattice simulation at (**p**) 1449 and (**q**) 2715 days. First incomplete moment is plotted against VAF. **r–t** Schematic illustration of boundary nucleation model at three sequential time points. **u** Fit of overall clone size distribution to power law. Overall, clone size distribution is plotted on a log–log scale. Line of best fit (grey), fit to 1/×2 (red) and fit to 1/×3 (blue) are shown

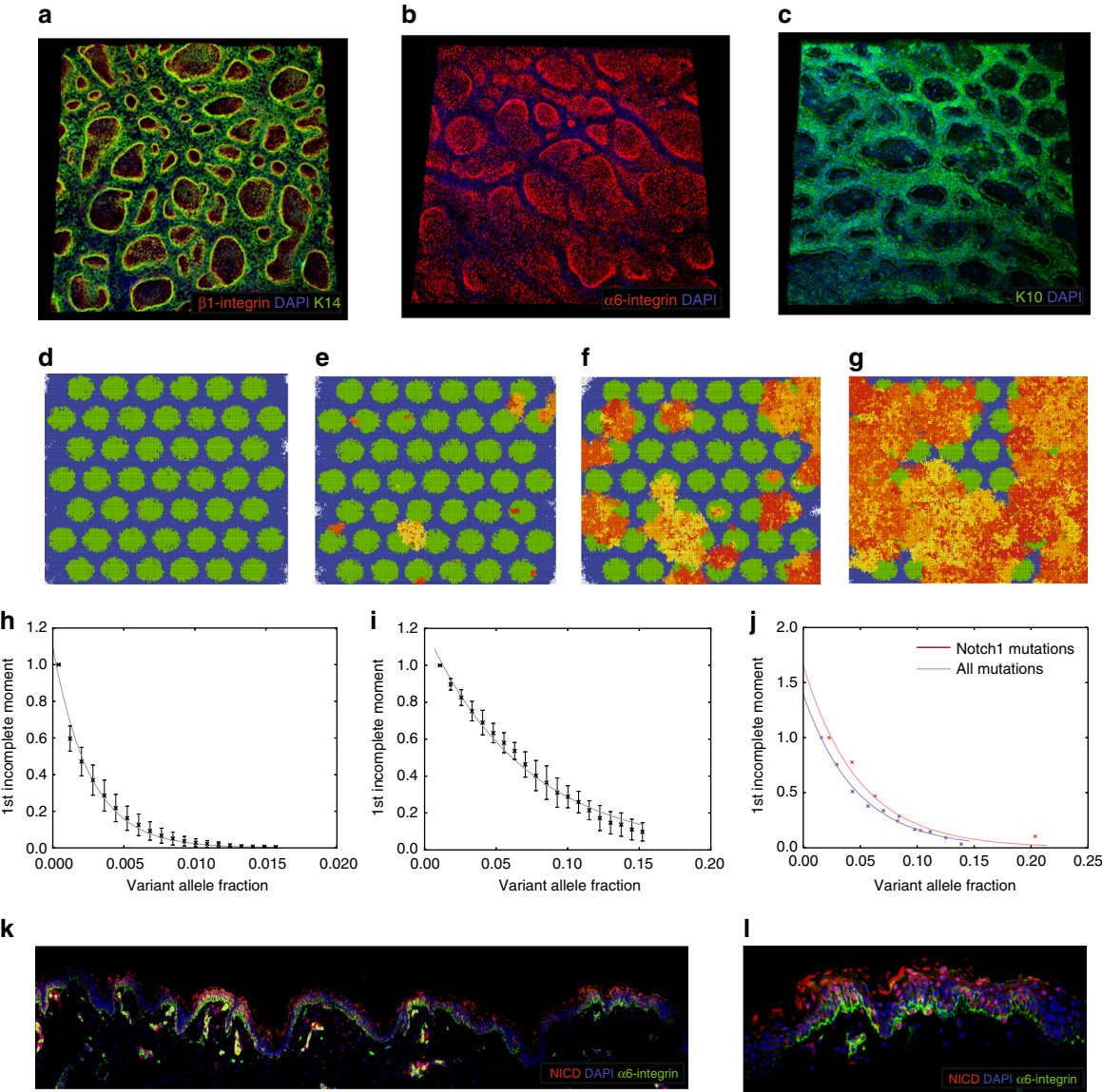

**Fig. 4** Modelling clonal expansion of mutations in the human epidermis. **a** Whole-mount imaging of basal layer of human epidermis with labelling of β1-integrin (red, stem cell marker), DAPI (blue, nuclear marker), Keratin 14 (green, basal layer marker). **b** Whole-mount imaging of basal layer of human epidermis with labelling of α6-integrin (red, stem cell marker) and DAPI (blue, nuclear marker). **c** Whole-mount labelling of basal layer of human epidermis with labelling of keratin 10 (green, differentiation marker) and DAPI (blue, nuclear marker). **d–g** Simulation of clonal evolution of 99,225 stem cells (green), transit amplifying cells (TA; blue) and empty spaces (white) on a 315 × 315 hexagonal lattice after 140 days (**f**) 620 days (**g**) 1100 days (**h**) and 1600 days (**i**). Stem cell cluster radius is 20 cells. Neutral mutation rate for all cells is $10^{-3}$. Notch mutations (yellow-red) arise stochastically with rate $10^{-6}$ cell$^{-1}$ day$^{-1}$ and prevent differentiation of stem cells to TA cells. Unique clones within an expanding Notch-mutant clone are indicated by an arbitrary colour within the yellow-red spectrum. **h** Clone size distribution resulting from simulation of stem cells and TA cells in the basal layer of the epidermis with neutral mutations only (10 years, 315 × 315 lattice, 10 replications with clones detectable by sequencing (>0.007 VAF) mean ± S.D.) **i** Clone size distribution resulting from simulation of stem cells and TA cells in the basal layer of the epidermis in the presence of Notch mutations. Note that scale is 10-fold greater than (**j**). (10 years, 315 × 315 lattice, 10 replications with clones detectable by sequencing (>0.007 VAF) mean ± S.D.). **j** Observed clone size distribution for all mutants (blue) and Notch1 mutants (red) detectable in the human epidermis. **k, l** Notch1 activity in the human epidermis. Active Notch1 signalling is indicated by positive staining for the Notch intracellular domain (NICD, red). α6-integrin (green, stem cell marker) and DAPI (blue, nuclear marker) are also indicated

Human epidermal whole-mounts were prepared and labelled using a procedure that was modified from previous reports[7, 43]. Breast or abdomen skin samples were obtained as surgical waste with appropriate ethical approval. Small pieces (2 cm²) of whole skin were treated with Dispase (Corning) overnight on ice at 4 °C. The epidermis was separated as an intact sheet from the dermis and immediately fixed in 4% paraformaldehyde for 1 h. Epidermal sheets were then washed and stored in PBS containing 0.2% sodium azide at 4 °C until staining. Sheets were permeabilised and blocked using PB buffer (0.5% skimmed milk powder, 0.25% fish skin gelatin, 0.5% Triton-X 100, HEPES-buffered saline) for 30 min at 37 °C in a shaking incubator at 100 r.p.m. Epidermal sheets were then stained with specific primary antibodies [anti-β1

integrin (clone P5D2; Millipore MAB1959; 1:500), FITC-conjugated anti-α6-integrin (CD49f) antibody (clone GOH3; 1:200), chicken anti-Krt14 (Covance, SIG2376, 1:500), rabbit anti-K10 (Covance, PRB-159P, 1:500)] in PB buffer overnight at 4 °C in a 24-well tissue culture plate. Samples were washed and incubated with corresponding secondary antibodies for 2 h in room temperature and washed before mounting on a slide with ProLong anti-fade mounting medium (Thermo Fisher) with DAPI (4′,6-diamidino-2-phenylindole) to reveal nuclei. A Nikon A1 confocal microscope was used for image acquisition. NIS Elements (Nikon Instruments Inc.) software was used for 3D maximal projection (1024 × 1024 dpi), volume rendering and deconvolution on z-stacked whole-mount images.

**Estimation of maximal clone size expected from neutral drift**. We have employed three approaches to understand the maximum clone sizes that are predicted to arise by neutral drift within the timespan of a human lifespan. Firstly, the size distribution of mutant clones of size $n$ at time $t$ assuming neutral drift is given by[12]:

$$P_n(t) = \frac{1}{\ln(r\lambda t)} \frac{e^{-n/r\lambda t}}{n},$$

where $r\lambda$ is the rate of stem cell loss/replacement in the basal layer. Thus, for a stem cell population of size $N$, the cumulative probability of a clone of size $s$ or greater is

$$C_n(t) = \sum_{n=s}^{N} \frac{1}{\ln(r\lambda t)} \frac{e^{-n/r\lambda t}}{n}$$

for a basal cell area of 4 mm$^2$ with an estimated basal cell progenitor cell density of 100 cell per mm, gives a total cell population of 400$^2$. Assuming a rate of loss/replacement $\lambda$ in the basal layer of 0.5 per cell per week, the probability of observing the largest clones in our data set, which comprise 64,000 cells (0.2 VAF, i.e., 40% of the cell population) gives a cumulative probability of $4.4 \times 10^{-13}$. Clearly, it is highly improbable that clones of this size will be observed in vivo.

Secondly, with analogy to a classical voter model of surface catalysis, Klein[44] derives the time to saturation $T$ for a lattice comprising two cell types in neutral competition ($r = 1/2$).

$$T = \frac{N\left(\ln N + \frac{1}{2r}\right)}{\lambda},$$

where $N = 64,000$ cells (our largest clones) the time to saturation was in excess of 29,000 years; a 1 mm$^2$ clone would require >2900 years according to this model.

Finally, we computationally simulated neutral drift on a 400 × 400 hexagonal lattice (see details below) for 100 years with $\lambda = 0.5$ week-1 with the stochastic occurrence of neutral mutations. The largest observed clones comprised a fraction of 0.025 of the total population (Supplementary Fig. 5i, j). So in summary, taken together, these approaches indicate that neutral drift would require at least an order of magnitude longer than a human lifespan to account for the clone sizes that we have observed.

In order to compare the clone size distribution observed in our data with the predictions of a model of neutral drift, we employed the formulation developed by Simons (Equation 2) *pmid*26699486. Simons provides a full derivation of this relationship in the supplementary text.

$$\frac{1}{\langle n(t)\rangle} \sum_{m=n}^{\infty} m P_m(t) \approx e^{-(n-n_0)/r\lambda t}$$

The left hand side represents the first incomplete moment and the the right hand side represents the exponential form assumed by this distribution, where a cellular population is subject to cell loss and replacement under the assumptions of neutral drift. Where $\frac{1}{\langle n(t)\rangle}$ represents the average mutant clone size, $n$ the mutant clone size, $r\lambda t$ a decay constant corresponding to the average size of a mutant clone induced at the time of first exposure to a mutation. In an identical manner to the plots presented by Simons, clone size distributions were fitted to two parameters: $r\lambda t$, and $n_0$—a correction for the size limit of high throughput sequencing for the detection of mutant clones.

**Derivation of geometric model of boundary nucleation**. We now describe a geometric formulation of the boundary nucleation model of clonal expansion. While this geometric formulation omits stochastic factors and other effects of clonal competition that are modelled by the lattice simulation, it does illustrate important concepts that govern the emergence of the observed clone size distribution.

We consider the case of a primary clone carrying a non-neutral mutation, which leads it to outcompete surrounding wild-type stem cells in a two-dimensional stem cell compartment such as the basal layer of the epidermis. We assume that $\omega \gg \theta$, where $\omega$ is the neutral mutation rate and $\theta$ the non-neutral mutation rate. A primary clone arises as a single cell and expands outwards as a circle. We assume that subsequent mutations are neutral and occur stochastically with rate $\omega$ for all cells in the stem cell compartment. Neutral mutations arising in wild-type cells are subject to neutral drift, similarly neutral mutations arising in mutant cells within the centre of the clone are also subject to neutral drift. However, if a neutral mutation occurs in a cell at the boundary of the mutant clone there is the possibility that this mutation will be amplified as a passenger along with the mutant clone as it expands displacing surrounding normal cells. As the size of the primary clone increases, the probability of nucleation of a secondary daughter clone containing an additional neutral mutation increases in proportion to the number of cells at the boundary; however, the potential size to which this secondary clone can expand decreases due to both competition with other cells at the boundary and since the primary clone is closer to its maximal size. Since clones arising via this mechanism are far larger than those arising by neutral drift within the same timescale, the latter are omitted from the model. This model is compatible

with both expansion terminated by collision between competing mutant clones or with an extrinsic process that limits expansion.

At time $T$, we observe a primary clone of radius $R > r \gg \sigma$, where $\sigma$ is the diameter of a single cell. This primary clone has expanded from a single cell in the period between $t_0$ and $T$. At time $t$, the radius of the primary clone is $r$ and the number of secondary clones $N$ that initiate at the boundary at $t$ is proportional to the number of cells that comprise the circumference,

$$N(t)\,\mathrm{d}t = \frac{2\pi r}{\sigma} \cdot \gamma \cdot \mathrm{d}t,$$

where $\gamma$ is a constant determined by the neutral mutations rate $\omega$ and the nature of the competitive advantage conferred unto the primary clone. Although some secondary clones will be extinguished and others will grow to larger size, on average, we assume that the area achieved by a secondary clone is given by

$$A(R, r) = \frac{\pi R^2 - \pi r^2}{2\pi r/\sigma} = \frac{R^2\sigma}{2r} - \frac{r\sigma}{2}.$$

It is noted that this model will not accurately represent the situation when $r \approx \sigma$, where stochastic factors will dominate and has no physical meaning where $2r < \sigma$. Rearranging to get $r$ and taking the positive-signed solution of the quadratic

$$-\sigma r^2 - 2rA + \sigma R^2 = 0$$

$$r = \sqrt{\left(\frac{A}{\sigma}\right)^2 + R^2} - \frac{A}{\sigma}$$

Integrating over time and changing variables gives $M$, the number of clones of size $A$ as a function of $A$

$$\int_0^T N(t)\,\mathrm{d}t = \int_{A_{max}}^0 N(t) \cdot \frac{\mathrm{d}t}{\mathrm{d}r} \cdot \frac{\mathrm{d}r}{\mathrm{d}A}\,\mathrm{d}A \equiv \int_{A_{max}}^0 M(A)\,\mathrm{d}A,$$

where $A_{max}$ is the size of the largest secondary clones. Taking the derivative of $r$ with respect to $A$

$$\frac{\mathrm{d}r}{\mathrm{d}A} = \frac{A}{\sigma^2 \sqrt{\left(\frac{A}{\sigma}\right)^2 + R^2}} - \frac{1}{\sigma}$$

We assume that the primary clone expands as $r(t) = \alpha t$ as a consequence of the competitive advantage of mutant over wild-type stem cells acting at the boundary. Substituting:

$$M(A) = \frac{2\pi\gamma}{\sigma} \cdot \frac{1}{\alpha} \cdot \left(\sqrt{\left(\frac{A}{\sigma}\right)^2 + R^2} - \frac{A}{\sigma}\right)\left(\frac{A}{\sigma^2 \sqrt{\left(\frac{A}{\sigma}\right)^2 + R^2}} - \frac{1}{\sigma}\right)$$

Our experimentally observed clone size distributions appear linear when plotted on a log–log scale (Fig. 3s). This is suggestive of an inverse power law distribution. Therefore, we anticipated an inverse power law dependence for our geometric model at large A. As for our experimental data, we can estimate the exponent, $\beta$, by taking the gradient of the logarithm of the distribution against the logarithm of area,

$$\beta(A) \equiv \frac{\mathrm{d}\ln(M(A))}{\mathrm{d}\ln(A)} = -\frac{A\left(A + 2\sqrt{A^2 + \sigma^2 R^2}\right)}{A^2 + \sigma^2 R^2}$$

taking the limit for large $A$ gives $\beta(\inf) = -3$. Thus, the distribution approaches an inverse cubic power law quickly as the size of secondary clones becomes non-negligible with regard to the primary clone. We can also derive this power law dependence from the leading term as $\sigma$ tends to a small number, keeping $R$ constant:

$$M(A) \sim \frac{2\pi\gamma}{\alpha} \cdot \frac{\delta R^4}{4A^3}$$

**Derivation of clone size distribution for exponential growth**. The lattice simulation and geometric model described above assume that all growth of the primary clone arises from division of cells at the boundary. As a consequence, the maximum rate at which the clone can expand is proportional to the number of cells at the boundary, i.e., the circumference of the clone. This gives a squared dependency for total clone size (area) on time. However, in some circumstances, it appears smaller clones can undergo a transient period of exponential growth[26].

For exponential growth, the rate of clonal expansion must be proportional to the total number of cells in the clone, implying that all cells in the clone have an equal probability of dividing. For a bacterial colony in suspension culture, this situation can arise since there is no spatial limitation on division, however, for a clone in a two-dimensional stem cell compartment, the maximum rate of growth is determined by the rate of flux of mutant cells into regions previously occupied by wild-type stem cells. This rate will be limited by the length of the boundary, the maximal rate of cellular migration and the rate at which neighbouring wild-type stem cells can be displaced. Since the total number of cells in the mutant clone

increases as radius squared, whereas the length of the boundary increases proportionate to the radius, this flux must inevitably become rate limiting and this may explain why small clones can show exponential growth initially but rapidly revert to a slower growth rate at larger sizes[26].

Assuming that mutant clones can expand exponentially in vivo, at least transiently, let us consider the implications for mutant clone size distributions assuming an analogous mechanism to that described for the boundary nucleation model. As above, we assume that $\omega \gg \theta$, where $\theta$ is the rate of non-neutral mutations that confer a competitive advantage and $\omega$ the rate of neutral mutations. It is straightforward to derive a model for the expected distribution of large (i.e., detectable by high throughput sequencing) mutations assuming exponential growth since both primary and secondary mutations must expand at the same rate and the rate of secondary clone initiation is proportional to the number of cells in the primary clone.

If we observe a primary clone $A^P$ containing a non-neutral mutation at time $T$ that has expanded exponentially to size $A^P_{\max}$, since time $t_0$ the area of the primary clone $A^P$ at time $t$ is given by

$$A^P(t) = \pi\left(\frac{\sigma}{2}\right)^2 e^{bt},$$

where $\sigma$ is the diameter of a single cell. From the time that they are nucleated, secondary clones must grow in proportion to the primary clone, and therefore at the final time $T$, the area of a secondary clone $A$ is determined by the time of nucleation $t$

$$A(t) = \pi\left(\frac{\sigma}{2}\right)^2 e^{b(T-t)}.$$

Rearranging

$$t = T - \frac{1}{b}\ln\left(\frac{4A}{\pi\sigma^2}\right)$$

We will need

$$\frac{dt}{dA} = -\frac{1}{bA}$$

The rate of new clone nucleation is proportional to the area of the primary clone

$$N(t) = \gamma a e^{bt}$$

So

$$M(A) = N(t) \cdot \frac{dt}{dA} = \frac{\gamma a \pi \sigma^2 e^{bT}}{4A} \frac{1}{bA},$$

which trivially has a power law dependence of inverse squared area.

**Lattice model.** The model is defined on a two-dimensional hexagonal lattice $L_i$ for which each site $i$ is occupied by either a wild-type stem cell denoted by $A_n$, where ($n$ denotes a clonal label) or is empty, denoted by $\emptyset$. We first consider the situation when wild-type stem cells are in neutral drift. In this scenario, the lattice $L$, is subject to the following transitions (Fig. 4a):

$$A_n \xrightarrow{r\lambda} A_n A_n$$
$$A_n \xrightarrow{r\lambda} \emptyset$$
$$A_n \rightarrow A_n P,$$

where $\rightarrow$ denotes a change in state of the lattice from $t_n$ to $t_{n+1}$, $r$ is a universal rate constant reflecting the period of chronological time that elapses from $t_n$ to $t_{n+1}$ and $\lambda$ is the rate of stem cell loss/replacement in the basal layer of the epidermis. $A_n \rightarrow A_n A_n$ denotes the replication of a stem cell to occupy two lattice positions. $P$ denotes a committed progenitor. $A_n \rightarrow \emptyset$ is equivalent to $A_n \rightarrow PP$ and indicates the loss of a stem cell from the basal layer via division into two commited progenitors. Since $A_n \rightarrow A_n P$ does not alter the number or position of stem cells in the lattice, it is not explicitly considered in the model.

$N_{i,t}$ is defined as the set of neighbours occupied by stem cells for position $i$ on the lattice at time $t$. The number of neighbours is given by $|N_{i,t}|$ and for a two-dimensional hexagonal lattice can assume a maximum value of 6. Individual members of the set are accessed via index $j$: $N_{i,t,j}$. For stem cells in neutral drift, a vacancy $\emptyset$ in the lattice has equal probability of being filled by each of its neighbours $N_{i,t}$:

$$\emptyset \rightarrow \begin{cases} N_{i,t,0} & \text{Prob.} \frac{1}{|N_{i,t}|} \\ \cdots & \\ N_{i,t,|N_{i,t}|-1} & \text{Prob.} \frac{1}{|N_{i,t}|} \end{cases}$$

At time $t_0$ all stem cells have clonal identity $A_0$. Subsequently, neutral mutations arise stochastically with rate $r\omega$, resulting in the assignment of a new, unique,

clonal label $m$ to cell $A$ and all descendents:

$$A_n \xrightarrow{r\omega} A_m$$

$\langle L \rangle_{n,t}$ is the number of cells on the lattice with clonal label $n$ at time $t$. $C_{n,t}$ is the set of clones (children) descended directly at from $A_n$ at time $t$ and all earlier time points. Thus, the clonal history of the model is represented by a tree with root node $A_0$ and the size of clone $n$ at time $t$ is given by a recursive sum of clone $n$ and all children of clone $n$ at time $t$:

$$\text{sum}(n,t) = \langle L \rangle_{n,t} + \sum_{c \in C_{n,t}} \text{sum}(c,t)$$

This methodology can represent clone bifurcation, whereby a subsets of the clone acquire different subsequent mutations and can thus accurately model the sequential accumulation of unique mutations within the genome.

Cellular migration on the lattice $L$ is implemented by stochastically swapping the contents of adjacent cells with rate $r\kappa$:

$$L_i \xrightarrow{r\kappa} L_j$$
$$L_j \xrightarrow{r\kappa} L_i,$$

where $L_{j,t} \in N_{i,t}$

Next, we consider the situation where mutations arise which alter neutral dynamics. In the context of the lattice model the most important of these is the probability that a stem cell is lost from $L$ with a rate different from $r\lambda$. Other factors include the possibility that a member of the set $N_{i,t}$ has a probability of filling a vacancy $\emptyset$, which differs from $\frac{1}{|N_{i,t}|}$ or that a cell $B_n$ alters the behaviour of other cells on the lattice (non-cell autonomous effects). Outside the context of this lattice model it is possible that mutant stem cells could alter the size of the stem cell compartment or altered mobility within the basal layer. For the purpose of this simulation, we are primarily interested in the fixation of secondary neutral mutations during expansion of the primary mutant clone and the precise nature of the replicative advantage is less important therefore we focus on the former scenario.

We allow non-neutral mutations to arise stochastically as follows:

$$A_n \xrightarrow{r\theta} B_m$$

A stem cell with non-neutral dynamics is denoted by $B$ and acquires a new clonal label $m$. In most circumstances, it is anticipated that $\theta \gg \omega$. The rate of cell loss from the lattice differs between cells with rate $r\lambda$ in cells of type $A$ and $r\phi$ for cells of type $B$. Thus, the lattice transitions governing the model of non-neutral dynamics are as follows:

$$A_n \xrightarrow{r\lambda} A_n A_n$$
$$A_n \xrightarrow{r\lambda} \emptyset$$
$$A_n \xrightarrow{r\omega} A_m$$
$$A_n \xrightarrow{r\theta} B_m$$
$$B_m \xrightarrow{r\lambda} B_m B_m$$
$$B_m \xrightarrow{r\phi} \emptyset$$
$$B_m \xrightarrow{r\omega} B_k$$

To further extend the model to a represent a heterogenous stem cell compartment containing both stem and TA cells, we introduce a novel cell type $C$ to represent a TA cell. $C$ is subject to the following lattice transitions:-

$$A_n \xrightarrow{r\nu} C_{n,0}$$
$$C_{n,q} \rightarrow C_{n,q+1} C_{n,q+1} \text{ if } q < q_{\max}$$
$$C_{n,q} \rightarrow \emptyset$$
$$C_{n,q} \xrightarrow{r\omega} C_{m,q},$$

where $r\nu$ is the rate of differentiation of stem cells into TA cells, $n$ is the clonal label associated with the cell, and $q$ is the total number of replications completed by the TA clone. $\nu$ is a function of location on the lattice $L_i$: $\nu = \text{diff}(i)$. While $\text{diff}(i)$ can take any form, for simplicity we constrain it to a binary output with biologically representative forms, whereby stem cells are grouped in clusters as in the basal layer of the human epidermis. To limit the proliferative potential of TA cells a maximum of $q_{max}$ replications are permitted.

Experimentally, we observe that stem cells and TA cells cluster together, furthermore TA cells are rarely seen within the TA cluster. To model this tendency, we define a lattice energy function $E$:

$$E(i) = \begin{cases} |N_i \cap C| + \text{diff}(i) \cdot p & \text{if } L_i \in A \\ |N_i \cap A| - |N_i \cap \emptyset| + (1 - \text{diff}(i)) \cdot p & \text{if } L_i \in C \end{cases},$$

where $A$ is the set of all stem cells, $C$ the set of all TA cells, $\emptyset$ the set of empty lattice spaces and $p$ an arbitrarily large scalar penalty. This function takes higher values,

where stem cells are admixed with TA cells or where TA cells are localised within an area that contains a stem cell cluster. For two neighbouring sites on the lattice, cellular migration is only permitted where this leads to an overall reduction in lattice energy.

The lattice models described above are not readily amenable to analytic approaches and therefore we proceeded to computational simulation.

**Computational simulation.** The lattice model was implemented as a computational simulation on a hexagonal lattice with dimensions 200 × 200 or 400 × 400 (Supplementary Methods). The rate of stem cell loss/replacement $r\lambda$ has been estimated as 0.5 per week[12] and the simulation was implemented with discrete quantised time intervals of 1 day. All lattice transitions were assumed to initiate and complete within a single time step. A large majority of new clones created are lost from the lattice as a result of neutral drift, and therefore, in order to prevent the memory requirements of the clonal history of the simulation from growing to an excessive size, it is necessary to periodically purge the clonal history of clones for which no members of that clone or its descendants survive.

**Code availability.** The model was implemented in C++ and Python. Source code is available at the following URL: https://github.com/latticesim/sim. With the exception of simulations taking an excessive amount of computational time, all simulations were run a minimum of 10 times with different random seeding and results are presented as mean ± standard deviation.

**Data availability.** A full list of identified mutations is provided (Supplementary Data 1). The sequencing data has been deposited in dbSNP under the code B151. All other remaining data are available within the Article and Supplementary Files, or available from the authors upon request.

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

## Acknowledgements

This work was funded by grants to F.M.W. from the UK Medical Research Council and Wellcome Trust and to M.D.L. from the Academy of Medical Sciences. We also gratefully acknowledge funding from the Department of Health via the National Institute for Health Research comprehensive Biomedical Research Centre award to Guy's & St

Thomas' National Health Service Foundation Trust in partnership with King?s College London and King's College Hospital NHS Foundation Trust. We are grateful to the genomics core facility at the Wellcome Trust Centre for Human Genetics, Oxford for assistance with sample preparation and sequencing.

## Author contributions

M.D.L., F.M.W., J.N.B., E.C. and R.M.: Designed the experiments. M.D.L. and K.L.-A.: Performed the experiments. M.D.L., C.N.S.L. and K.L.-A.: Analysed the data. M.L.-A. and F.M.W.: Wrote the manuscript.

## Additional information

**Competing interests:** The authors declare no competing financial interests.

