## [Peer Review File · Nature Communications]

REVIEWERS' COMMENTS:

Reviewer #1 (Remarks to the Author):

The main novelty of this manuscript resides within the competition aspects, and not in the genetic approaches or the scaling law that the authors describe. We don't believe that experimental manipulations are within the scope of this manuscript, as feasibility seems low. If the authors did some mouse experiments with NOTCH1 or other genes unearthed by their human sequencing such as P53, RAI1, and PLXNB3, then the paper would likely be of great impact, but the timeframe of those experiments is long, and this seems beyond what is reasonable to demand of the authors.

In assessing the manuscript based on Review 1's original comments on the unrevised manuscript, we believe the major contribution of this manuscript is with regard to cell competition: mutations at the edge of a clone have a competitive advantage over those at the middle of a clone. Additionally, the authors show that large clones could not be generated through neutral drift alone. In contrast to Reviewer 1, we believe that the modeling approaches of the authors do not constitute pure description. Rather they are a valid form of validation of the data, by varying parameters. The authors make these observations through previously unattained numbers and sizes of human samples.

While Reviewer 1 misunderstood the definition of universal scaling laws, the scaling law mentioned in the title seems of concern for a different reason, namely that it is only measured under three orders of magnitude. The "universal scaling law" should be removed from the title.

The genomic sequencing approaches reported do not represent significant advances in this field, and the thrust of the manuscript is not in the discovery of novel genetic regulators. Nevertheless, the manuscript still makes a contribution that would be of interest to the scientific community, particularly in the fields of cell competition and human genetics, if the paper could be reframed to emphasize geometrical factors and cell competition, as Reviewer 3 has already pointed out.

Additional comment: The K10+ cells marked in the basal layer (Figure 4C) are confusing, as K10 typically marks suprabasal cells. Can the authors clarify?

Minor note: nearly all of the figures are extremely hard to read because the labels are tiny (likely far below the minimum font size for any journal). The labels must be reformatted prior to publication.

Reviewer #2 commented for the editors only and was satisfied with the revised manuscript.

REVIEWERS' COMMENTS:

Reviewer #1 (Remarks to the Author):

The main novelty of this manuscript resides within the competition aspects, and not in the genetic approaches or the scaling law that the authors describe. We don't believe that experimental manipulations are within the scope of this manuscript, as feasibility seems low. If the authors did some mouse experiments with NOTCH1 or other genes unearthed by their human sequencing such as P53, RAI1, and PLXNB3, then the paper would likely be of great impact, but the timeframe of those experiments is long, and this seems beyond what is reasonable to demand of the authors.

In assessing the manuscript based on Review 1's original comments on the unrevised manuscript, we believe the major contribution of this manuscript is with regard to cell competition: mutations at the edge of a clone have a competitive advantage over those at the middle of a clone. Additionally, the authors show that large clones could not be generated through neutral drift alone. In contrast to Reviewer 1, we believe that the modeling approaches of the authors do not constitute pure description. Rather they are a valid form of validation of the data, by varying parameters. The authors make these observations through previously unattained numbers and sizes of human samples.

While Reviewer 1 misunderstood the definition of universal scaling laws, the scaling law mentioned in the title seems of concern for a different reason, namely that it is only measured under three orders of magnitude. The "universal scaling law" should be removed from the title.

We have changed the title to remove the concept of a universal scaling law and emphasize the cell competition that arises from spatial constraints.

The genomic sequencing approaches reported do not represent significant advances in this field, and the thrust of the manuscript is not in the discovery of novel genetic regulators. Nevertheless, the manuscript still makes a contribution that would be of interest to the scientific community, particularly in the fields of cell competition and human genetics, if the paper could be reframed to emphasize geometrical factors and cell competition, as Reviewer 3 has already pointed out.

We have modified the paper to highlight the work in the context of cell competition. This includes alterations to the title, abstract, introduction and conclusions to emphasize relevance to this field.

Additional comment: The K10+ cells marked in the basal layer (Figure 4C) are confusing, as K10 typically marks suprabasal cells. Can the authors clarify?

In the basal layer of the epidermis 'islands' of beta-1-integrin high stem cells are surrounded by scattered cells that express K10. These cells typically have a mushroom shape as they are in the process of exiting the basal layer. K10 is, as the reviewer notes, predominantly expressed in suprabasal cells. This has been previously published e.g. Fig 4 in Jensen, Development 126:2409.

Minor note: nearly all of the figures are extremely hard to read because the labels are tiny (likely far below the minimum font size for any journal). The labels must be reformatted prior to publication.

We have modified the figures to increase the size of labels in order to increase readability.

Reviewer #2 commented for the editors only and was satisfied with the revised manuscript.